# A Novel High-Speed Resonant Frequency Tracking Method Using Transient Characteristics in a Piezoelectric Transducer

**DOI:** 10.3390/s22176378

**Published:** 2022-08-24

**Authors:** Jeonghoon Moon, Sungjun Park, Sangkil Lim

**Affiliations:** 1DH Innovation Co., Ltd., Gwangju 61209, Korea; 2Department of Electrical Engineering, University of Chonnam National, Gwangju 61186, Korea; 3Department of Automotive Engineering, University of Honam, Gwangju 62399, Korea

**Keywords:** resonant frequency tracking, piezoelectric transducer, ultrasonic, curve fitting, underdamped response characteristics

## Abstract

When driving the piezoelectric transducer (PT: piezo transducer), which is a key device, it is important for the ultrasonic system (using ultrasonic waves of 20 kHz or higher) to operate at a resonant frequency that can maximize the conversion of mechanical energy (vibration) from electrical energy. The resonant frequency of the PT changes during the actual operation according to the load fluctuations and environmental conditions. Therefore, to maintain a stable output in an ultrasonic system, it is essential to track the resonant frequency in a short time. In particular, fast resonant frequency tracking (RFT: resonant frequency tracking) is an important factor in the medical ultrasonic system, i.e., the system applied in this thesis. The reason is that in the case of a medical ultrasonic system, heat-induced skin necrosis, etc., may cause the procedure to be completed within a short period of time. Therefore, tracking the RFT time for maximum power transfer is an important factor; in this thesis, we propose a new high-speed RFT method. The proposed method finds the whole system resonance frequency by using the transient phenomenon (underdamped response characteristic) that appears in an impedance system, such as an ultrasonic generator, and uses this to derive the mechanical resonance frequency of the PT. To increase the accuracy of the proposed method, parameter fluctuations of the pressure of the PT, the equivalent circuit impedance analysis of the PT, and a MATLAB simulation were performed. Through this, the correlation between the resonance frequency of the ultrasonic system, including the LC filter with nonlinear characteristics and the mechanical resonance frequency of the PT, was analyzed. Based on the analyzed results, a method for tracking the mechanical resonance frequency that can transfer the maximum output to the PT is proposed in this thesis. Experiments show that using the proposed high-speed RFT method, the ultrasonic system can track the mechanical resonance frequency of the PT with high accuracy in a short time.

## 1. Introduction

Ultrasonic systems are used for various purposes, such as semiconductor material processing, biological tissue cutting, and ultrasonic welding, by converting electrical energy into microscopic vibrations (mechanical energy), and amplifying them [1,2]. The principle of the ultrasonic system is that heating, welding, and cutting are performed through the combined action of heat generated by stress concentration and heat generated by vibration displacement (vibration speed, amplitude) [3]. The configuration of the ultrasonic system can be largely divided into an electrical oscillation unit and a mechanical oscillation unit [4]. The mechanical oscillation part consists of a vibrator called the bolt-clamped Langevin-type transducer (BLT), in which the piezo element is bolted, and there is a booster to amplify the mechanical displacement to fit the object to be joined (and to avoid mechanical displacement by contacting the object to be joined). It consists of a horn that transmits to the junction [5,6,7]. The key device that generates ultrasonic waves is called a piezo transducer (PT: piezo transducer), and the PT includes a piezo element that converts electrical energy into vibration, which is mechanical energy [8,9]. In general, the equivalent model representation of the PT is expressed as an impedance model composed of the RLC series circuit part as well as the parallel capacitor *C* that directly affect the mechanical output [10]. The PT that is represented by the impedance model must be operated at a resonant frequency, in which the impedance is minimum to maximize the transfer from electrical energy to mechanical energy [11,12]. However, the resonant frequency fluctuates because the parameter value fluctuates by external conditions, such as the load, pressure, temperature, and operating state [13,14]. The electrical oscillator involves an inverter generating a sine wave voltage of 20 kHz or higher to induce the vibration of the PT; this is called an ultrasonic generator [15,16].

Ultrasonic generators require several functions for high-quality energy transfer. Firstly, a frequency control function is required because the output frequency must be operated as a resonant frequency in order to minimize loss due to reactive power and increase energy density. As mentioned above, the power converter requires a fast RFT for a stable output because the parameter value fluctuates depending on external conditions [17]. An important point is that the series RLC part configured in the PT directly affects the mechanical output, so it is necessary to track the resonance frequency [18]. At this time, the resonance frequency is called the mechanical resonance frequency. In addition, since the actual operation process times of many ultrasonic systems are short (seconds), the RFT time is an important factor [19,20]. Among them, in the case of the medical ultrasonic system to be applied in this thesis, it can be said that the high-speed RFT is even more important because it can cause problems (e.g., skin necrosis) due to heat [21]. As described above, since the RFT method in the ultrasonic system has many factors to consider, related research is being actively conducted via various methods. Methods include tracking parallel resonance frequency [9], maximum power [12,20,22], maximum admittance [23], maintaining a constant amplitude [24,25,26,27,28], etc.; various methods have been attempted and implemented. Although these methods have obtained relatively good results, most have complex structures and problems that can only be applied to a specific system or field. In addition, even if the output power is sufficiently high, the PT is not excited at the mechanical resonance frequency of the PT, so there is a problem in that the life of the PT is shortened in the ultrasonic system. Moreover, in the case of the phase lock loop (PLL), which is a general frequency control method, it is an RFT method in which soft switching can be performed from the point of view of a power converter without considering the effects of a parallel capacitor configured in a filter circuit and a PT in an ultrasonic generator [27,29,30,31,32].

Accordingly, in this thesis, we developed a high-speed RFT method that can quickly track the mechanical resonant frequency ($f_{piezo}$), which can transfer the maximum mechanical output and avoid the effects of parallel capacitors and matching circuits. The proposed high-speed RFT method can increase the accuracy by tracking the mechanical resonance frequency using the measurable whole system resonance frequency; the principle is as follows. Due to *R* and *LC* of the PT configured at the output stage of the ultrasonic system, an output with an underdamped characteristic appears when a pulse voltage is applied, and the zero-point period of this output becomes the whole system resonance frequency ($f_{system}$). In addition, the process of measuring the ultrasonic system resonance frequency by detecting the period of the zero-point of the current has the advantage of being able to perform the ultrasonic system RFT at a high speed because only a circuit that detects the zero-point of the current needs to be configured without a separate operation. However, the correlation between the ultrasonic system resonance frequency and the mechanical resonance frequency of the PT has a non-linear characteristic; the correlation was analyzed according to the change in the external conditions of the PT. Based on the analysis data and results, a high-speed tracking algorithm for mechanical resonance frequency that can generate maximum power to the PT through curve fitting was proposed, and this was verified through simulations and experiments. The composition of the thesis is as follows. Section 2 examines the characteristics of the PT model and the resonance frequency characteristics. Moreover, the characteristics when combined with the PT and the LC filter configured in the ultrasonic generator are examined. Section 3 describes the existing PLL method and the RFT method proposed in this thesis. Section 4 analyzes the characteristics of the PT and presents simulations for the control. Section 5 presents the experiments to verify the fast RFT method proposed in this thesis; it also presents the conclusions.

## 2. Analysis of the Piezoelectric Transducer

### 2.1. Equivalent Model and Resonant Characteristics

The PT can be expressed as an equivalent model as shown in Figure 1. The arm on the left in the figure is the electrical part containing the parallel capacitor $C_{0}$. The other arm is a mechanical part consisting of a dynamic capacitor $C_{1}$, a dynamic inductor $L_{1}$, and a dynamic resistance $R_{1}$, which directly affects the mechanical output. As can be seen from the equivalent model, the PT has a structure with two resonance points as shown in the following equation [20,30,33].
(1)$$f_{r}=\frac{1}{2\pi \sqrt{L_{1}C_{1}}}$$
(2)$$f_{a}=\frac{1}{2\pi \sqrt{L_{1}\left(\frac{C_{1}C_{0}}{C_{1}+C_{0}}\right)}}$$

The resonance frequency ($f_{r}$) of the RLC series circuit of the mechanical arm is arranged as in Equation (Equation 1), and this becomes the mechanical resonance frequency that transfers the maximum power to the PT. Moreover, the whole resonance frequency ($f_{a}$) of the PT, including the parallel capacitor $C_{0}$, is arranged as in Equation (Equation 2). That is, the PT has a resonant frequency and an anti-resonant frequency under the influence of the parallel capacitor $C_{0}$ [34].

The admittance graph of the PT is represented in Figure 2; Table 1 shows the characteristics for each frequency. At this time, if the loss resistance and load are neglected at each frequency, it can be expressed as $f_{m}$ = $f_{s}$ = $f_{r}$ and $f_{n}$ = $f_{p}$ = $f_{a}$ [35,36]. However, the loss resistance and load are affected in an actual ultrasonic system, so the condition for the maximum power transmission is the maximum mechanical series resonance frequency $f_{s}$ of conductance. Therefore, it is important to track the mechanical resonance frequency ($f_{s}$) and apply it to the system.

With the PT, the resonant frequency fluctuates because the parameter value fluctuates according to external conditions, such as the load, pressure, temperature, and operating state [13,14]. However, in the case of piezoelectric elements, the internal parameter values appear differently depending on the material, shape, and configuration, so the parameters are generally checked through measurements using a measuring instrument. Figure 3 shows the experimental data to look for the characteristics of the internal parameters according to the pressure conditions of the outer diameter (15 mm), inner diameter (6 mm), 2 T, and four-stacked PT applied in this thesis. The acquired data represent the average values through repeated tests 10 or more times to increase reliability. As a result of analyzing the data, it can be inferred that the mechanical resonance frequency will continue to fluctuate because the parameter value of the PT fluctuates due to the pressure change. However, it can be seen that the parallel capacitor $C_{0}$ value has a small change in the parameter according to the pressure change compared to other parameters. As a result, it can be inferred that the variation of the resonant frequency due to the value of the parallel capacitor $C_{0}$ will be small compared to the variation of the resonant frequency due to the series RLC value. Moreover, if the system is designed to ignore the effect of the PT parallel capacitor $C_{0}$ when designing the LC filter of the inverter output stage in the ultrasonic system, the resonance frequency fluctuation of the PT will be smaller. As a result, the whole equivalent model of the ultrasonic system can be newly analyzed with the RLC series circuit part and the LC filter part that affect the mechanical output of the PT [35,37]. This will be further explained in the following Section 2.2.

### 2.2. Analysis of the Whole System Equivalent Model

Figure 4 is the whole system equivalent circuit of the inverter output stage in the ultrasonic system. *L* and *C* configured in the LC filter are referred to as $L_{f}$ and $C_{f}$, respectively. Since the parallel capacitor $C_{f}$ and the capacitor $C_{f}$ are connected in parallel, it can be expressed as one capacitor $C_{2}$ as shown in Figure 5. Accordingly, if the $C_{f}$ value of the filter is designed high enough to ignore the effect of the parallel capacitor $C_{f}$, it will be possible to separate the RLC series circuit impedance and the LC filter impedance that affect the mechanical output of the PT as shown in Figure 5 [30,38,39].

Figure 6 shows the data obtained through the actual test mentioned in Figure 3 and the result value when the value of $C_{f}$ is increased 10 times than that of the value of $C_{0}$ and equalized with $C_{2}$. It can be predicted that the variation of the parallel capacitor $C_{0}$ value of the PT will have little effect on the resonant frequency of the PT, and that the effect will be smaller as $C_{f}$ is increased.

To analyze the whole system impedance, each impedance $Z_{1}$, $Z_{2}$, $Z_{p}$ is shown in Figure 7, and the equation for this is as follows.

$L_{1}$$L_{f}$ reactance and $C_{1}$, $C_{2}$ reactance is as follows:(3)$$\begin{array}{c} X_{L1}=wL_{1},X_{C1}=-\frac{1}{wC_{1}}\\ X_{Lf}=wL_{f},X_{C2}=-\frac{1}{wC_{2}} \end{array}$$

PT $R_{1}$ + $L_{1}$ + $C_{1}$ total $Z_{p}$ impedance is as follows:(4)$$\begin{array}{c} Z_{p}=R_{1}+jX_{L1}+jX_{C1}\\ =R_{1}+j\left(X_{L1}+X_{C1}\right) \end{array}$$

$Z_{1}$ parallel impedance, excluding the filter $L_{f}$ impedance, is as follows:(5)$$\begin{array}{cc} Z_{1} & =\frac{Z_{p}\times jX_{C2}}{Z_{p}+jX_{C2}}\\ \; & =\frac{\left(R_{1}+j\left(X_{L1}+X_{C1}\right)\right)\times jX_{C2}}{\left(R_{1}+j\left(X_{L1}+X_{C1}\right)\right)+jX_{C2}}\\ \; & =\frac{-X_{L1}X_{C2}-X_{C1}X_{C2}+jRX_{C2}}{R+j(X_{L1}+X_{C1}+X_{C2})} \end{array}$$

$Z_{p}$ is the series RLC impedance of the PT and $Z_{1}$ can be expressed as Equation (Equation 5) above due to $C_{2}$. Through this, the whole system impedance $Z_{2}$, including the inductor $L_{f}$ of the LC filter, can be expressed as Equation (Equation 6) below.

Total equivalent impedance $Z_{2}$, including the inductor $L_{f}$ of the LC filter, is as follows:(6)$$\begin{array}{cc} Z_{2} & =\frac{-X_{L1}X_{C2}-X_{C1}X_{C2}+jRX_{C2}}{R+j(X_{L1}+X_{C1}+X_{C2})}+jX_{Lf} \end{array}$$

As shown in Equation (Equation 6), when the $L_{f}$ and $C_{f}$ parameter values of the LC filter are determined and the series RLC parameter values of the PT are determined, the resonance impedance of the whole system, including the LC filter and the PT, can be analyzed [40,41]. As in Section 2.1, resonant frequency characteristics of PT, the mechanical resonance frequency of the PT does not have a linear characteristic due to the fluctuation of the $L_{1}$ and $C_{1}$ parameter values of the PT. As a result, it was determined that an algorithm that can track the non-linear PT mechanical resonance frequency is needed based on the data on the ultrasonic system resonance frequency that can be obtained as a result of the analysis of the simulation conducted in the later chapter. Therefore, it was attempted to track the mechanical resonance frequency of the PT using MATLAB curve fitting.

## 3. Resonant Tracking Method

### 3.1. Conventional Resonant Tracking Method

In general, a PLL technique is widely used as a method for tracking a resonant frequency. Figure 8 and Figure 9 are the block diagram of the RFT method (using PLL among the existing methods for tracking the resonant frequency of a PT) and a graph of the phase change over time [42,43]. The existing method applies a voltage and current to the inverter, determines the polarity, brings a phase difference, and uses a frequency variable method by a proportional integral controller (PI controller) to make the phase difference 0 [44,45]. However, in the system in which the parameters of the PT change rapidly, including the medical system applied in this thesis, the PLL method has several disadvantages. Firstly, it takes several tens to hundreds of ms to track the resonance frequency due to the characteristics of the PI controller and the transient characteristics of *L* and *C* when tracking the resonance frequency. For this reason, this method is not suitable for systems in which the resonant frequency continuously changes rapidly due to changes in external conditions (pressure, temperature, etc.). Moreover, this method performs RFT for the purpose of soft switching, such as zero current switching (ZCS) and zero voltage switching (ZVS), to improve the efficiency of the whole power converter system [46,47]. However, in a system that requires intermittent and instantaneous output, such as a general ultrasonic system, or a system with a small power capacity, efficiency is not considered an important factor [48]. For the above reasons, the RFT method using the conventional PLL in the PT is not suitable.

### 3.2. Proposed Resonant Tracking Method

In case a voltage is initially applied to an ultrasonic system, including an LC circuit, a transient phenomenon will occur due to the exchange of magnetic energy and electrostatic energy of the inductor and capacitor configured in the circuit [49,50]. This can be derived from the characteristic equation representing the intrinsic characteristics of the real system, and the natural response characteristic can be confirmed through the root derived from the characteristic equation, which is called the characteristic root [51]. The characteristic root determines the response characteristics of the circuit according to the values of the damping frequency and the resonant frequency [52]. In general, the characteristic of PT is that the resonant frequency is greater than the damping frequency, so the underdamped characteristic appears; the resonant frequency was tracked through this characteristic [53]. In this case, the resonant frequency means the whole system’s resonant frequency. The characteristic equation of the equivalent circuit shown in Figure 5 is as follows in Equation (Equation 7).
(7)$$\begin{array}{cc} Z_{2} & \left(s\right)=\frac{C_{0}C_{1}L_{0}L_{1}\left(C_{1}L_{1}s^{2}+C_{1}R_{1}s+1\right)}{C_{0}C_{1}L_{0}L_{1}s^{4}+C_{0}C_{1}L_{0}R_{1}s^{3}+\left(C_{0}L_{0}+C_{1}L_{0}+C_{1}L_{1}\right)s^{2}+C_{1}R_{1}s+1} \end{array}$$

It is complicated (and difficult) to obtain the characteristic root of the characteristic equation of the fourth-order system as above [54], the response characteristic of the circuit can be analyzed through the s-plane, which is shown in Figure 10.

As shown in Figure 10 above, poles with different real number values and frequency characteristics exist in the s-plane by the transfer function of the fourth-order system. At this time, if the real number (a2) of the pole, which is far away from the imaginary part of the two poles, is 5 to 10 times larger than the size of the real part (a1) of the pole close to the imaginary axis; it can be divided into a dominant pole and an insignificant pole [55,56]. In general, the transient response characteristics of insignificant poles show fast decay and the dominant poles have relatively slow decay [57]; the characteristics are shown in Figure 11. Using these characteristics, we attempted to detect the whole resonant frequency of the circuit applied in this thesis and find the mechanical resonant frequency of the PT through this.

The proposed RFT method is summarized as follows. In case a pulse-type voltage is applied to a circuit composed of *L* and *C*, a transient phenomenon in the form of an underdamped characteristic waveform occurs due to the exchange of electrostatic energy and magnetic energy of the inductor and capacitor [58]. At this time, the transient phenomenon of the fourth system applied in this system shows attenuation characteristics and resonance frequency due to the dominant and insignificant poles [59]. After all the insignificant poles with relatively fast damping characteristics are attenuated, only the damping characteristics and the resonance frequency by the dominant pole exist, and the resonance frequency means the resonance frequency by *L* and *C* of the whole system at this time. By detecting the zero-point of the current through a high-speed analog circuit, the whole system resonance frequency was tracked, and through this, the mechanical resonance frequency of the PT was found. The correlation between the resonant frequency of the whole system and the mechanical resonant frequency of the PT can be analyzed through Equation (Equation 6) mentioned above; it can be expected that nonlinear characteristics will appear due to the characteristics of the circuit where *L* and *C* exist. Accordingly, it was attempted to increase the degree of tracking through curve fitting.

Figure 12 and Figure 13 are the block diagram of the proposed method for tracking the resonant frequency of the PT and the graph of the phase change with time. A gate circuit that can apply a square wave [60], which is an initial pulse-type voltage, and a comparator circuit [61] that can generate a pulse width modulation (PWM) signal by detecting the zero-point of the current generated after the square wave is applied were constructed [62,63]. For this reason, compared to the existing method, it is possible to significantly reduce the controller calculation time for RFT, and there is an advantage that only the frequency detection circuit needs to be configured. Tracking the resonance frequency using the above method is also the whole resonance frequency of the system ($f_{system}$), not the mechanical resonance frequency of the PT ($f_{piezo}$). The resonant frequency of the whole system was detected using a frequency detection circuit. After that, the new frequency based on the curve fitting data, in other words, the mechanical resonance frequency of the PT, was calculated, and the finally calculated frequency was used to perform PWM switching ($f_{sw}$) of the inverter through the microcontroller unit (MCU). As a result, an algorithm that detects the whole system frequency information and tracks the mechanical resonance frequency that can transfer the maximum power to the PT in the shortest time—even under the condition that the mechanical resonance frequency of the PT changes rapidly due to irregular and continuous external conditions—and an A curve-fitting analysis to analyze a nonlinear model [64] are proposed in this thesis.

Figure 14 shows the control algorithm for tracking the mechanical resonance frequency of the PT using the ultrasonic system resonance frequency. The control algorithm can be divided into three main modes, and each mode is controlled according to the mode switching ($f_{sw}$) of the MCU [65]. The initial operation is a section in which the resonant frequency based on the designed data of the PT [66,67] is directly controlled by PWM hard-switching through the MCU [68,69]. At this time, a transient occurs in the ultrasonic system, and the whole system’s resonant frequency can also be detected. After that, the whole system resonance frequency ($f_{system}$) is detected using a frequency detection circuit through mode switching. Using the whole system resonance frequency detected during this section, a new frequency based on the curve fitting data, in other words, the mechanical resonance frequency of the PT ($f_{piezo}$), is calculated. Finally, a control algorithm is configured to PWM switching ($f_{sw}$) of the calculated mechanical resonance frequency of the PT again through the MCU.

## 4. Simulation

A simulation was performed to confirm the resonance frequency characteristics according to the pressure fluctuation of the PT. The simulation conditions for each parameter are as follows. The minimum and maximum parameter fluctuation conditions of $L_{1}$ and $C_{1}$ were set as the parameter fluctuation values for the pressure fluctuation range, 0 to 5 kg (the actual use area of the medical PT). The fluctuation conditions were set for the values of parameters, $L_{1}$ and $C_{1}$, which affected the resonance frequency fluctuation of the PT in a monotonically increasing pattern from the minimum value. For the parallel capacitor $C_{0}$, the average value of 3.25 nF according to the pressure fluctuation for 0 to 5 kg of the medical PT used in the experiment of this paper was applied. In general, the capacitor $C_{f}$ value of the LC filter was selected to be about 10 times larger than the value of the parallel capacitor $C_{0}$ of the PT, so 33 nF was selected. For the inductor $L_{f}$, a general capacitance value of 870 μH was selected [70].

The above conditions of the simulation are summarized and presented in Table 2 below.

Figure 15a shows the results of the PT resonance frequency fluctuation when the $C_{1}$ value is set to the minimum and maximum values and the $L_{1}$ value is continuously varied from 190 to 240 mH. At this time, the average value of the resonant frequency of the PT was 30.2 kHz, indicating the simulation result. Figure 15b shows the results when the resonance frequency of the PT is shown when the $L_{1}$ value is set to the minimum, maximum, and intermediate values of 190, 215, 240 mH, and the $C_{1}$ value is continuously changed from 120 to 140 pF. Similarly, it was shown that the average value of the resonant frequency of the PT was 30.2 kHz. In general, for impedance matching in an ultrasonic system, the LC filter resonant frequency value is similarly designed by comparing it with the average resonant frequency of the PT [33,71]. For this reason, in the system of this paper, the LC filter resonant frequency value, including the PT parallel $C_{0}$, was finally designed to be 30.2 kHz, and simulations and experiments were conducted. Moreover, as shown in the above simulation results, the mechanical resonance frequency of the PT had a non-linear characteristic rather than a linear characteristic due to the parameter change of $L_{1}$ and $C_{1}$. Accordingly, the curve fitting simulation was performed to analyze the nonlinear characteristics of the whole system resonance frequency and the mechanical resonance frequency of the PT.

Figure 16 shows the simulation results of the correlation between the system resonance frequency and the PT resonance frequency. As shown in the figure, it can be seen that the whole system resonance frequency of the x-axis and the resonance frequency of the PT on the y-axis are not the same frequency. This means that in an ultrasonic generator, the whole system resonant frequency is not the mechanical resonant frequency that transfers the maximum power to the PT. If the control algorithm proposed in the above chapter is used, the whole system resonance frequency, which is the frequency on the x-axis, can be detected, and a new frequency on the y-axis, i.e., the mechanical resonance frequency of the PT, can be calculated by the curve fitting data. In summary, it is possible to find out the frequency of the y-axis by detecting the frequency of the x-axis. However, in the PT, the frequencies, as shown in Table 1 in Section 2, are actually in positions that are close to each other. Accordingly, accurate frequency tracking is required to transfer maximum power to the PT. Therefore, it was judged that information other than the whole system resonance frequency was needed for accurate frequency tracking. Accordingly, the correlation was analyzed between the mechanical resonance frequency of the PT and the inverter phase difference theta.

Figure 17 shows the simulation results of the correlation between the mechanical resonance frequency of the PT and the inverter phase difference theta. It can be confirmed that the inverter phase difference theta of the x-axis, with respect to the mechanical resonance frequency of the PT on the y-axis, also has a non-linear characteristic. As shown in the simulation results, if the mechanical resonance frequency of the PT is traced using the information of the phase difference theta of the inverter as well as the system resonance frequency, more accurate frequency tracking will be possible.

Figure 18 is a 3D simulation result showing the correlation between the system resonance frequency, the inverter phase difference theta, and the mechanical resonance frequency of the PT; the overall aspect of each correlation can be seen.

As shown in the simulation results, each correlation is shown to have non-linear characteristics. In addition, if the system resonance frequency and inverter phase difference theta are used, the PT resonance frequency can be accurately tracked according to each correlation. Accordingly, the mechanical resonance frequency of the PT was accurately tracked by detecting the inverter phase difference theta and the system resonance frequency as shown in the simulation results using curve fitting suitable for analyzing the nonlinear model.

## 5. Experiment Result

In this section, the correlation between the system resonance frequency and the mechanical resonance frequency of the PT was checked using the data according to the pressure fluctuations of the medical PT. Moreover, the experimental waveform of the ultrasonic generator to which the resonance tracking algorithm proposed in this paper is applied as shown.

Figure 19 shows the experimental environment and configuration. It consists of a control board, a power board, a PT, and a press system for pressure fluctuations of the PT. The press system for pressure fluctuations was designed in a structure that could apply pressure in the vertical direction to detect the pressure using a strain gauge to create a condition that could always apply constant pressure, and to analyze the fluctuations of the procedure to a minimum.

Figure 20 shows the experimental results, showing the mechanical resonance frequency of the PT based on the serial RLC parameter value of the measured PT when the pressure applied to the medical PT varied from 0 to 5 kg in units of 250 g. It can be seen that the mechanical resonance frequency of the PT appeared non-linearly according to the pressure fluctuation.

Figure 21 is the result of expressing the mechanical resonance frequency of the PT measured in the pressure fluctuation experiment applied to the medical PT as red dot data in the simulation result of the correlation between the system resonance frequency and the mechanical resonance frequency of the PT. Accordingly, when using the actual medical PT, it is possible to know the range correlation between the whole range of the mechanical resonance frequency of the PT and the system resonance frequency. From the above experimental result, it can be confirmed that the range of the mechanical resonance frequency of the PT for the actual use area of the medical PT is located in the range of 30.7 to 31.6 kHz of the system resonance frequency of the x-axis. As a result, if the PT resonant frequency value of the y-axis is calculated and tracked for the system resonant frequency of the x-axis, the mechanical resonant frequency that can transfer the maximum power to the PT can be tracked. In addition, the section other than the red dot data can be seen as a section that is out of the pressure fluctuation Section (0 to 5 kg), which is the actual use area of the medical PT. In other words, if the control algorithm is applied using the curve fitting data to the red dot data section, which is the actual area of use based on the experimental results, the mechanical resonance frequency that is able to transfer the maximum power to the PT on the y-axis can be tracked according to the system resonance frequency on the x-axis is calculated.

The following shows the experimental waveform of the ultrasonic generator to which the resonance tracking algorithm proposed in this paper is applied. Firstly, the result was confirmed based on the voltage and current of the inverter output stage as the point for detecting the resonance frequency of the system. According to the sequence of the control algorithm in the previous section, mode 1 is for the initial operation step that applies PWM switching to the designed data resonance frequency of the PT, and the next step is to calculate the mechanical resonance frequency of the PT by detecting the system resonance frequency using the frequency detection circuit. This step in which the algorithm is applied is called mode 2, and the final step of operating at the mechanical resonance frequency of the last calculated PT is called mode 3; the classification of each mode was confirmed using a trigger signal.

Figure 22 shows the resulting waveform when changing from mode 1 to mode 2 to check the system RFT. It can be seen that the voltage and current phases of the inverter are out of phase when PWM switching is operated at 32 kHz in mode 1. Moreover, it can be seen that the resonant frequency of the system is tracked as soon as the mode is changed to mode 2, which is the time when the trigger is low.

Figure 23 is the resulting waveform at the time of changing from mode 2 to which the PT mechanical resonance frequency calculation algorithm (based on curve fitting data) is applied to mode 3, which switches to the calculated frequency. In mode 2, it can be confirmed that the voltage and current phases of the inverter continue to match until the PWM switching operation is finished, and it is confirmed that the switching frequency at this time is 31.3 kHz.

Figure 24 is a waveform for confirming that the mechanical resonance frequency of the PT calculated by the curve fitting data based on the system resonance frequency detected in mode 2 switches normally in mode 3. The operating waveform of the mechanical resonance frequency of the PT in mode 3 was confirmed based on the system resonance frequency 31.3 kHz detected in mode 2, and the resonance frequency of the PT was set at 30.4 kHz when compared with the curve fitting data in Figure 25. It was confirmed that the tracking was performed normally. The following is the result of the experimental waveform to which the proposed RFT algorithm was applied at the input stage of the PT. In the same manner as in the previous operation, the three-step state of the control algorithm was checked.

Figure 26 shows the timing of the PWM switching operation at the initial 32 kHz in mode 1 and the operation in mode 2, and it can be seen that the phase difference between the voltage and current of the PT occurs in both modes.

Figure 27 shows the transition time from mode 2 to mode 3. In mode 2, the voltage and current phases of the PT almost coincide at the point where the system resonant frequency detection is almost finished through the frequency detection circuit, but this is a characteristic that occurs in a transient state and it is not the actual mechanical resonant frequency of the PT.

Figure 28 is a waveform for confirming that the mechanical resonance frequency of the PT calculated by the curve fitting data based on the system resonance frequency detected in mode 2 switches normally in mode 3. The operating waveform of the mechanical resonance frequency of the PT in mode 3 was confirmed based on the system resonance frequency 31.2 kHz detected in mode 2, and the mechanical resonance frequency of the PT was 30.4 kHz when compared with the curve fitting data in Figure 29. It was confirmed that the tracking was performed normally. According to the above experimental results, we verified the feasibility of tracing the mechanical resonance frequency that can transfer maximum power to the PT by using the curve fitting data and applying the proposed algorithm.

## 6. Conclusions

In this thesis, we proposed an algorithm that can track the resonant frequency at a high speed by using the transient (underdamped characteristic) that appears in an impedance system, such as an ultrasonic generator. The validity of the proposed algorithm was verified through simulations and experiments, and it can be summarized as follows.

1. In the process of tracing the resonant frequency, which is a disadvantage of the general RFT method, the problem of it taking several tens to hundreds of ms (due to the calculation process of the controller) has been improved. To this end, the whole system resonance frequency was detected using the zero-point of the transient response current generated by the resonance characteristic in the circuit composed of *L* and *C*, and the resonance frequency tracking speed was increased by removing the delay time by the PI controller.

2. To track the mechanical resonance frequency for transferring the maximum power to the PT, an equivalent model analysis of the PT and the resonance characteristics were analyzed. In addition, the system was designed by analyzing the impedance of the series RLC circuit that affects the mechanical output of the LC filter and the PT. Through this, the correlation between the system resonance frequency, the inverter phase difference theta, and the mechanical resonance frequency of the PT was analyzed and a control algorithm using curve fitting data was applied. As a result, it was possible to detect the resonance frequency of the whole ultrasonic generator system and track the mechanical resonance frequency for transferring the maximum power to the actual PT using the inverter phase difference theta information.

3. It was confirmed that the RFT calculation time takes about 400 μs as a result of the experiment of the system to which the proposed algorithm is applied using curve fitting data. Through this, it was verified that the algorithm is suitable for tracking the resonance frequency in the shortest time even under the condition that the mechanical resonance frequency of the PT changes rapidly.

Therefore, the algorithm proposed in this thesis improves the disadvantages of the existing RFT methods of PT, and it is judged to be useful in a system in which the mechanical resonant frequency of a PT changes rapidly due to irregular and continuous fluctuations depending on external conditions.

## Figures and Tables

**Figure 1 sensors-22-06378-f001:**
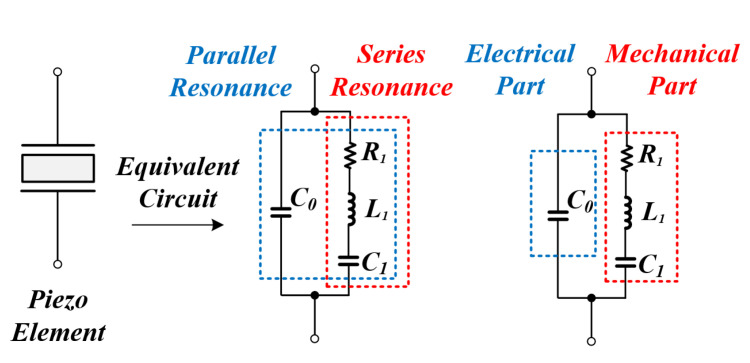
PT equivalent model.

**Figure 2 sensors-22-06378-f002:**
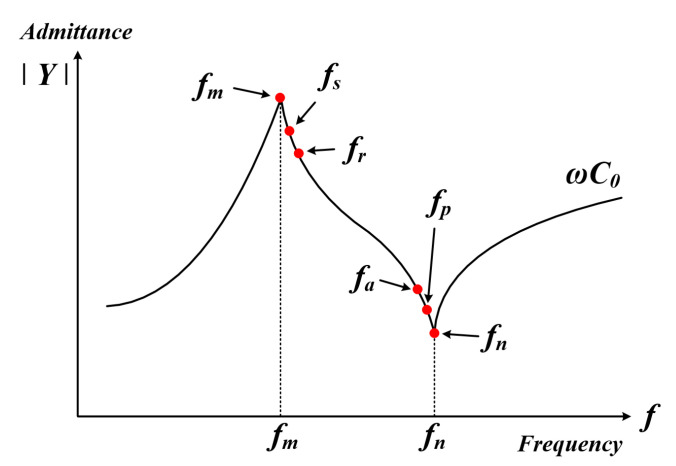
Admittance graph of the PT.

**Figure 3 sensors-22-06378-f003:**
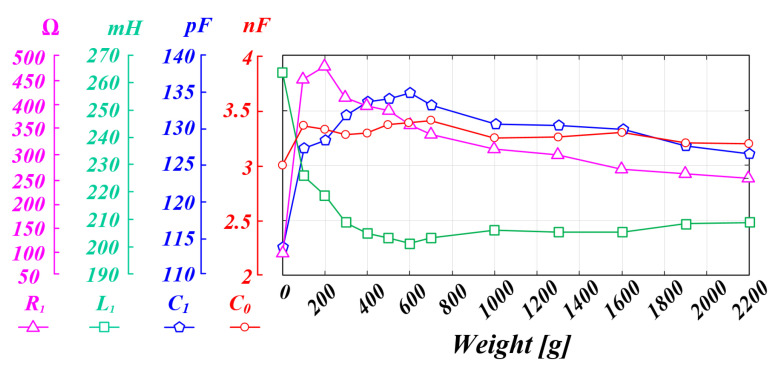
PT parameter change characteristic graph according to pressure fluctuation.

**Figure 4 sensors-22-06378-f004:**
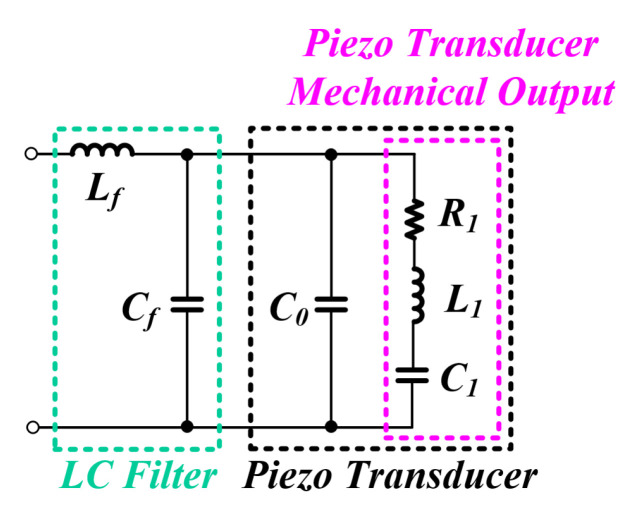
Whole system equivalent model.

**Figure 5 sensors-22-06378-f005:**
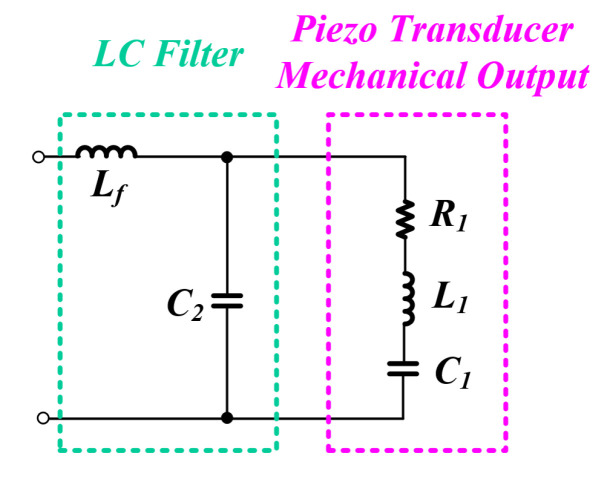
Impedance separation of the LC filter and PT RLC series circuit.

**Figure 6 sensors-22-06378-f006:**
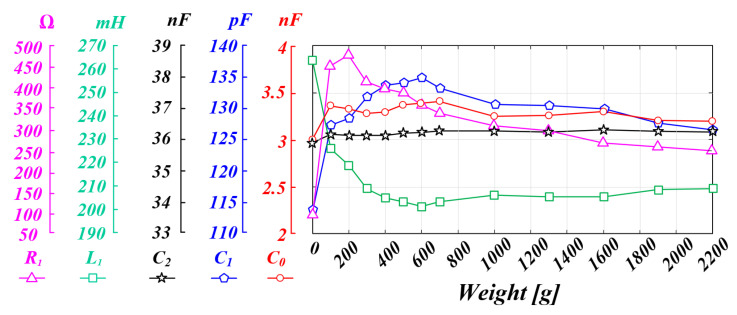
PT parameter change characteristic graph according to pressure fluctuation 2.

**Figure 7 sensors-22-06378-f007:**
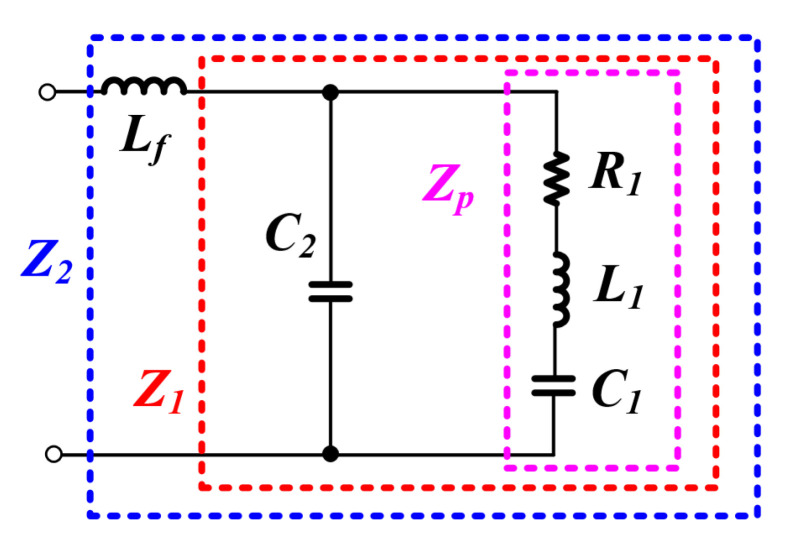
Impedance $Z_{1}$, $Z_{2}$, $Z_{p}$ equivalent model.

**Figure 8 sensors-22-06378-f008:**
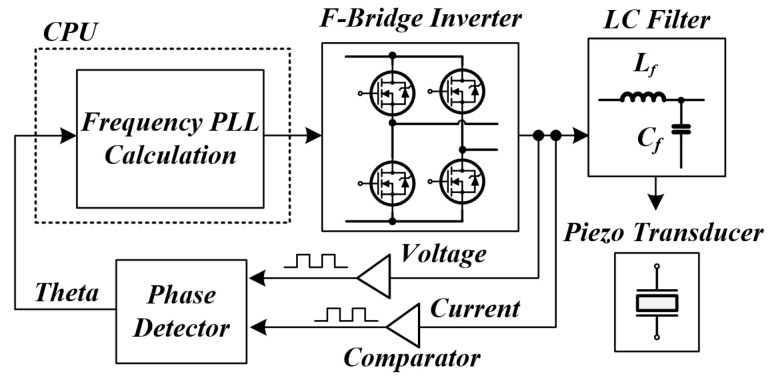
Block diagram of conventional resonance tracking method.

**Figure 9 sensors-22-06378-f009:**
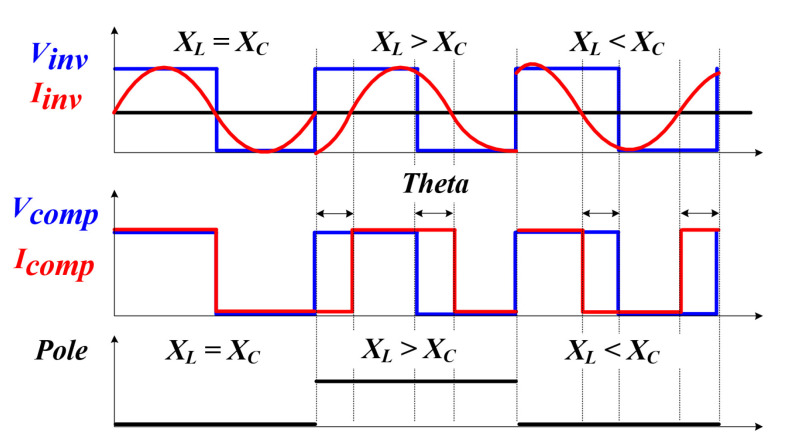
Phase graph of conventional resonance tracking method according to the time change.

**Figure 10 sensors-22-06378-f010:**
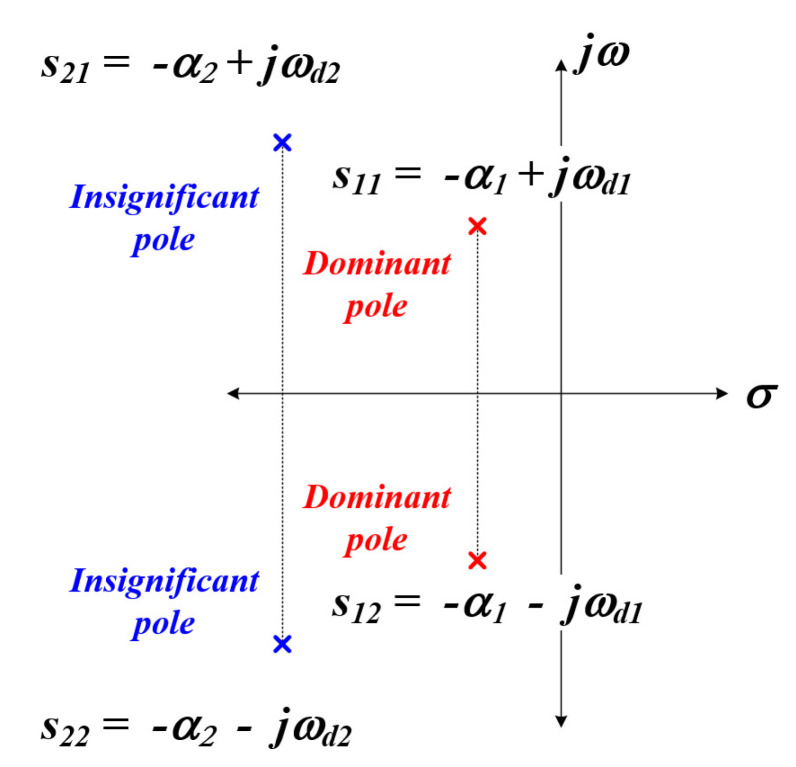
Fourth system s-plane.

**Figure 11 sensors-22-06378-f011:**
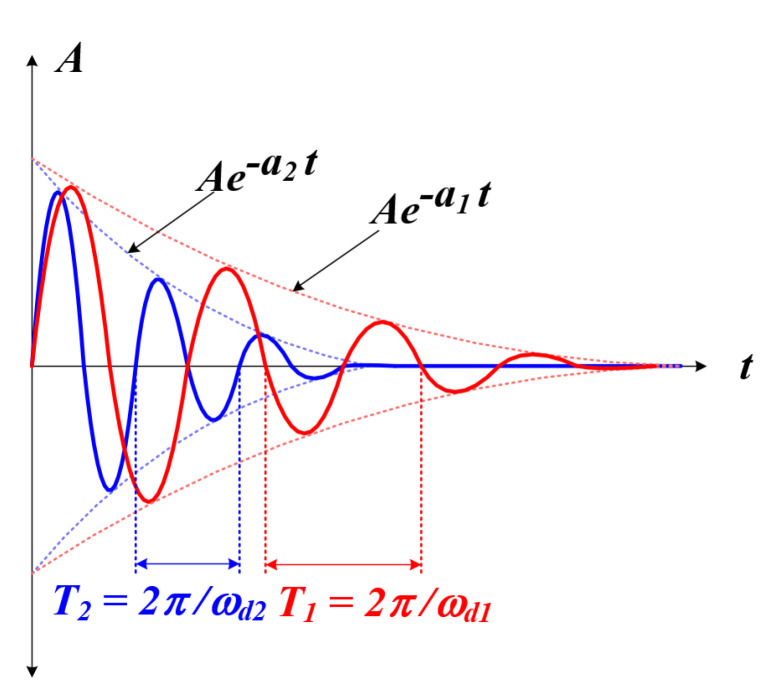
Fourth system response characteristics.

**Figure 12 sensors-22-06378-f012:**
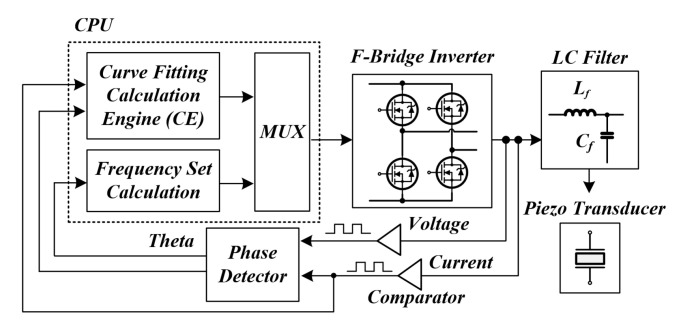
Block diagram of the proposed resonance tracking method.

**Figure 13 sensors-22-06378-f013:**
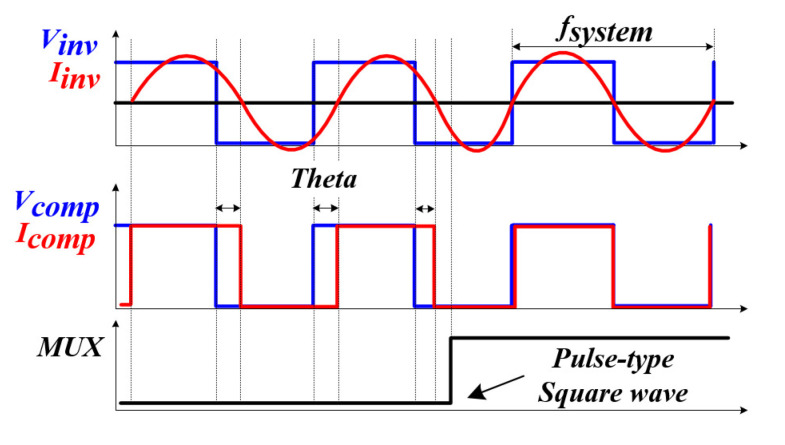
Phase graph of the proposed resonance tracking method according to the time change.

**Figure 14 sensors-22-06378-f014:**
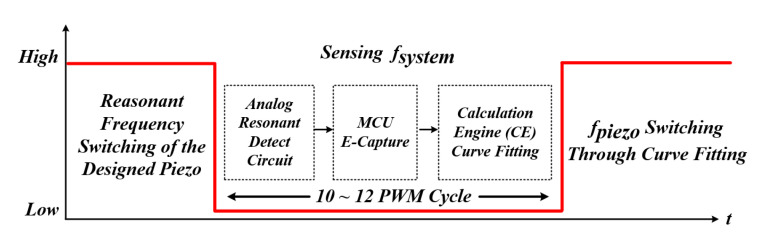
Control algorithm.

**Figure 15 sensors-22-06378-f015:**
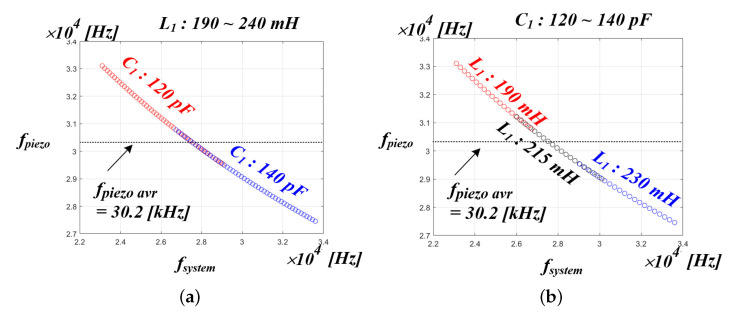
Range of PT resonance frequency according to $L_{1}$ and $C_{1}$ fluctuations for (**a**) $C_{1}$—120, 140 pF and the $L_{1}$—190 to 240 mH and (**b**) $L_{1}$—190, 215, 240 mH, and the $C_{1}$—120 to 140 pF.

**Figure 16 sensors-22-06378-f016:**
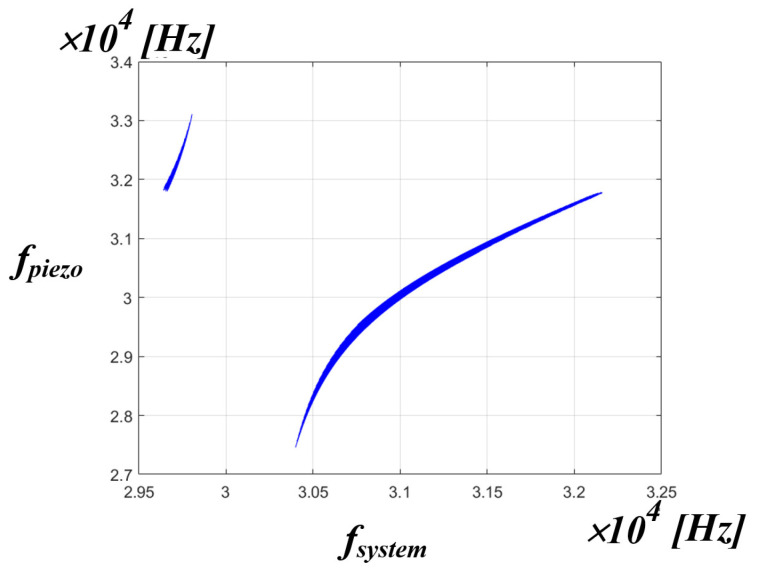
Simulation result of the correlation between the $f_{system}$ and $f_{piezo}$.

**Figure 17 sensors-22-06378-f017:**
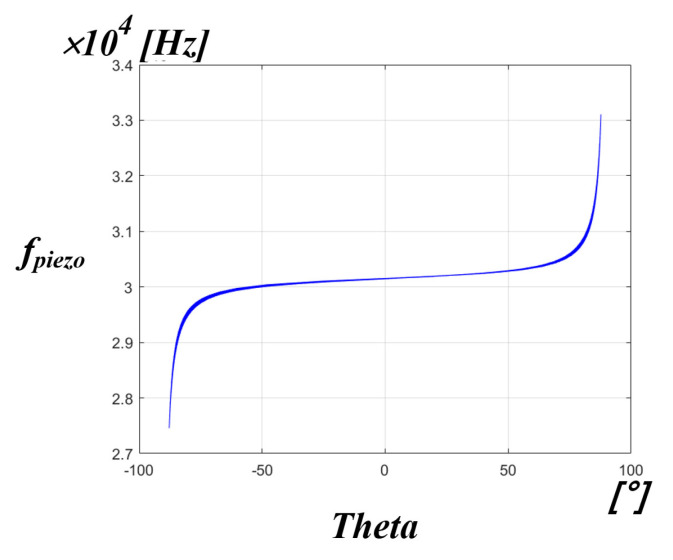
Simulation result of the correlation between the $f_{piezo}$ and the inverter phase difference theta.

**Figure 18 sensors-22-06378-f018:**
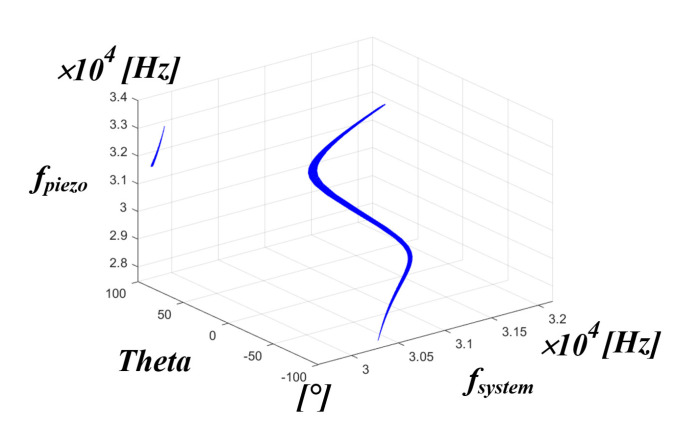
The 3D simulation result of $f_{system}$, inverter phase difference theta, and $f_{piezo}$.

**Figure 19 sensors-22-06378-f019:**
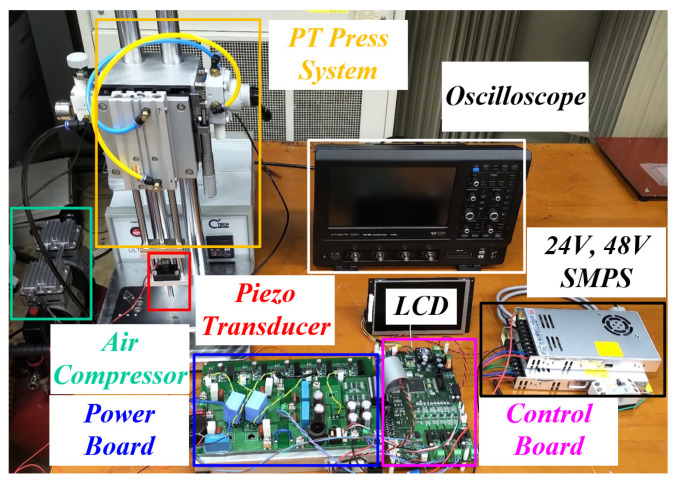
Experimental environment and configuration.

**Figure 20 sensors-22-06378-f020:**
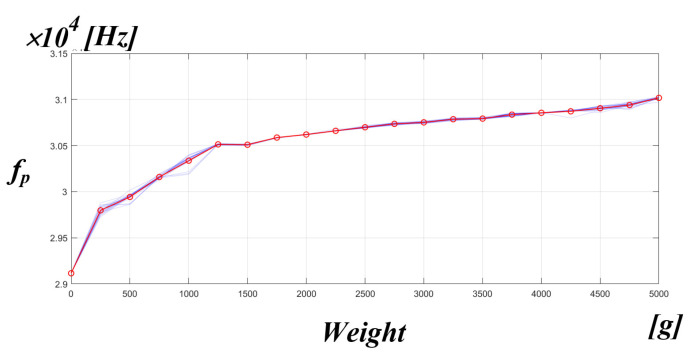
$f_{piezo}$ fluctuations due to pressure fluctuations.

**Figure 21 sensors-22-06378-f021:**
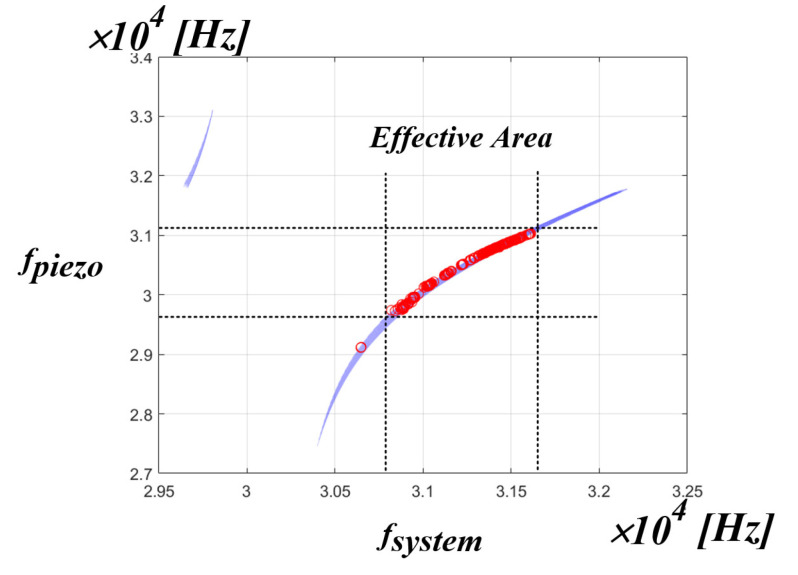
$f_{piezo}$ distribution area.

**Figure 22 sensors-22-06378-f022:**
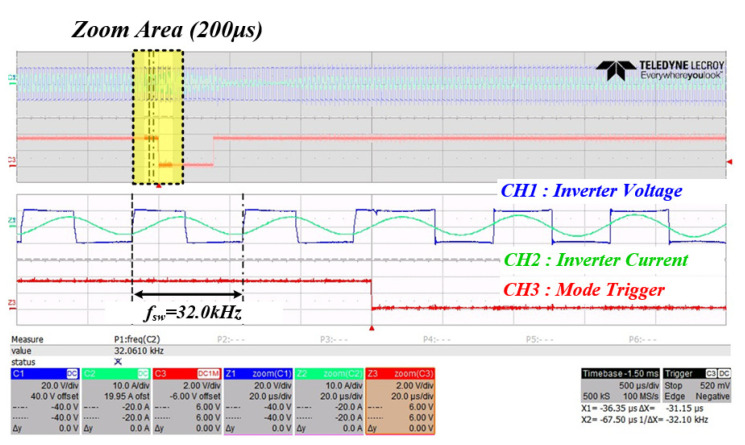
Inverter output waveform when mode 1 changes to mode 2 (CH1: inverter voltage (blue), CH2: inverter current (green), CH3: mode trigger (red)).

**Figure 23 sensors-22-06378-f023:**
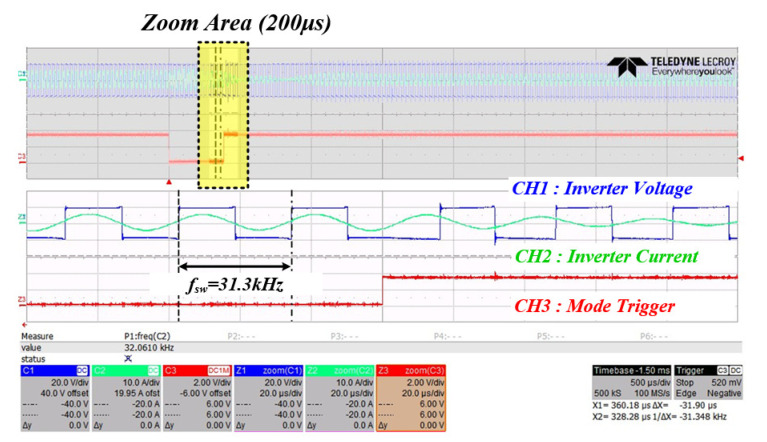
Inverter output waveform when mode 2 changes to mode 3 (CH1: inverter voltage (blue), CH2: inverter current (green), CH3: mode trigger (red)).

**Figure 24 sensors-22-06378-f024:**
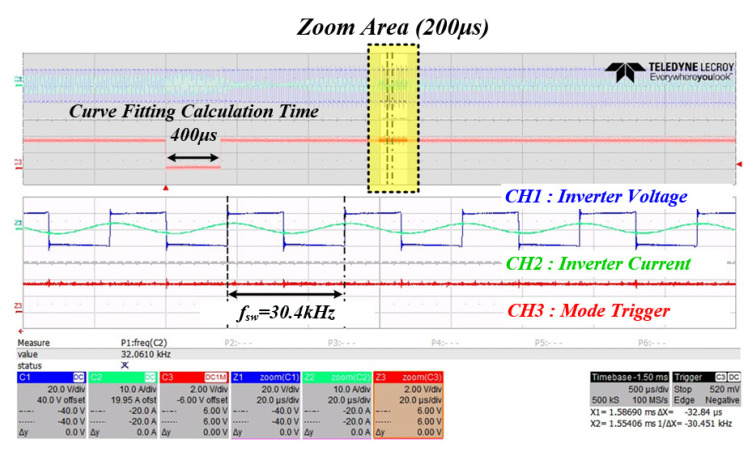
Inverter output waveform when in mode 3.

**Figure 25 sensors-22-06378-f025:**
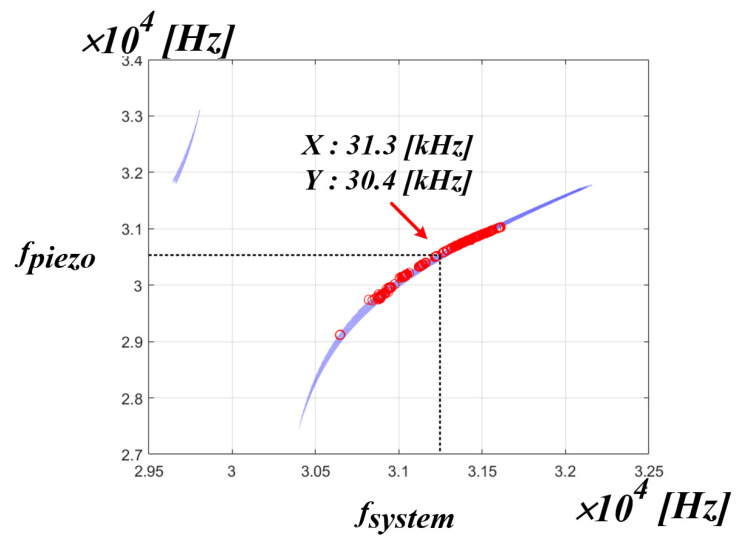
Curve fitting data 1.

**Figure 26 sensors-22-06378-f026:**
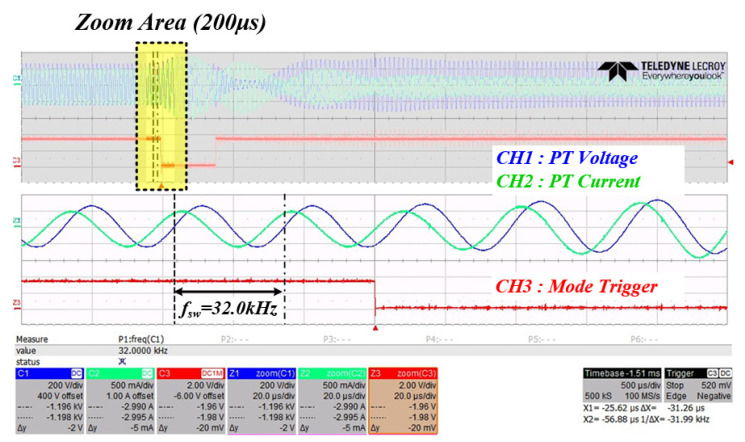
PT output waveform when mode 1 changes to mode 2 (CH1: PT voltage (blue), CH2: PT current (green), CH3: mode trigger (red)).

**Figure 27 sensors-22-06378-f027:**
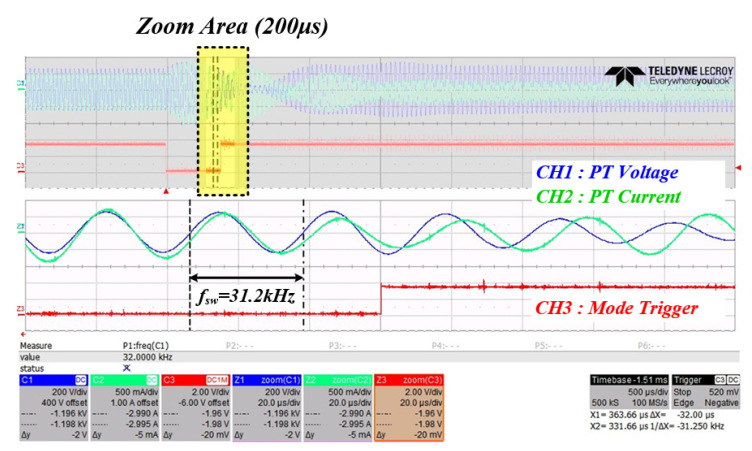
PT output waveform when mode 2 changes to mode 3 (CH1: PT voltage (blue), CH2: PT current (green), CH3: mode trigger (red)).

**Figure 28 sensors-22-06378-f028:**
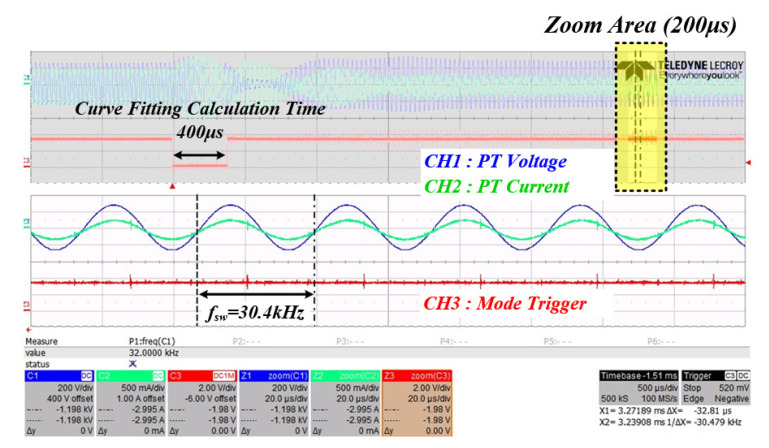
PT output waveform when in mode 3.

**Figure 29 sensors-22-06378-f029:**
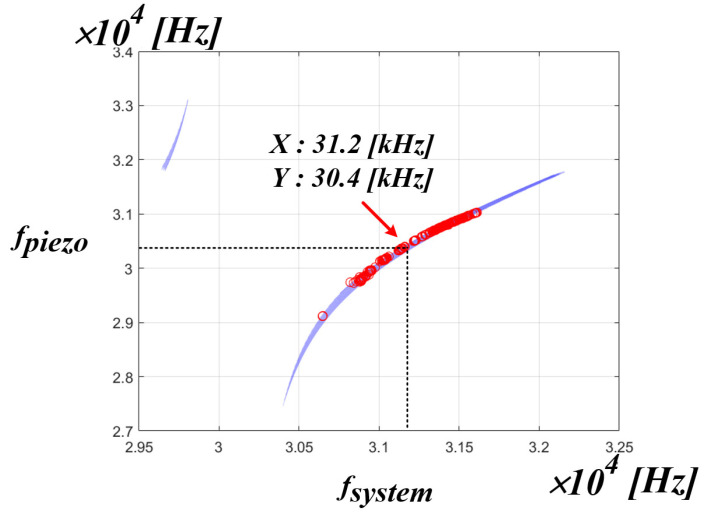
Curve fitting data 2.

**Table 1 sensors-22-06378-t001:** Frequency characteristics of the PT.

Frequency	Characteristics
$f_{s}$	Mechanical series resonant frequency
$f_{p}$	Mechanical parallel resonant frequency
$f_{r}$	Resonant frequency
$f_{a}$	Anti-resonant frequency
$f_{m}$	Admittance absolute value becomes extremely large
$f_{n}$	Admittance absolute value becomes extremely small

**Table 2 sensors-22-06378-t002:** Practical application areas of medical piezo element MATLAB simulation conditions.

Parameters	Parameter Values	Conditions
$R_{1}$	220 Ω	-
$L_{1}$	190 to 240 mH	Increase by 1 mH
$C_{1}$	120 to 140 pF	Increase by 1 pF
$C_{0}$	3.25 nF	Fix values
$L_{f}$	870 μH	-
$C_{f}$	33 nF	-
LC Filter resonant frequency	28.3 kHz	Filter capacitor $C_{2}$

## Data Availability

Not applicable.

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
