# Peer review of "A Novel High-Speed Resonant Frequency Tracking Method Using Transient Characteristics in a Piezoelectric Transducer"

_sensors, 2022, doi:10.3390/s22176378_

Round 1
Reviewer 1 Report
This manuscript proposes a new method to fast Resonance frequency tracking (RFT) of a piezoelectric transducer. The main idea of the method is tracking the entire system resonance frequency by detecting the zero point of current through a high-speed analog circuit. The authors give a detailed rational, though, with a little wordy, on the relationship between the mechanical resonance frequency and the entire system resonance frequency at the transient when a pulse-type voltage is beginning applied to the piezoelectric transducer. To my understanding, they considered that there exists at this nearly zero-point time a transient phenomenon in the form of under-braking occurring due to the exchange of electrostatic energy and magnetic energy of the inductor and capacitor. Among the transient phenomenon of the fourth-order system in the formal of equation (7) of this piezoelectric transducer system includes the to-be-tracking resonance frequency and it would show attenuation characteristics. Then, follows the building of the detect circuit and algorithm to arrive at their positive results, through experiments and simulations.
Nevertheless, there are some limitations or problems among the text-writing, listed as follows.
(1) The present title seems too redundant and not correctly reflect the novelty of their study, instead, it may suggest to use the following modification: A novel method for high-speed tracking resonant frequency of piezoelectric transducer. And accordingly, the whole text may write briefly further.
(2) How get the data in Fig.3, Fig.4, Fig.6?
(3) Is the fa in equation (1) equal to that of appearing in Table 1? And where shows the meaning of f1 and f2?
(4) In section 3.1, should cite a literature for the PLL technique?
(5) In equation (7), missing the right part at the equal-sign?
(6) Line 41-45, Line 54-57, Line 83-85, etc., redundant, and not a good writing.
Reviewer 2 Report
Although the paper would seem to be really interesting, unfortunately it is extremely difficult to read and follow.
I would like to pay attention to the following problems.
In the abstract, it is not clear why resonant frequency detection for a piezoelectric transducer is a relevant problem. This is a fundamental input for the reader, and, for such reason, it should be carefully defined.
The introduction is too concise and the aforementioned studies on resonance frequency tracking should be better described (e.g. lines 48-64) to give the reader a suitable overview. Furthermore, some fundamental concepts are not accurately explained (e.g. lines 25-27).
In Section 2 there are many shortcomings:
- the models of piezoelectric elements, although widely described, are not supported by the references to validate the considerations made;
- the language used should be revised to correct repetitions and reduce the complexity of the text;
-I do not understand where figures 3 and 6 come from. Are they graphs of the results of the paper?
-what is Section A of Chapter 2? I cannot find it.
In Section 3 some knowledge, even if basic to the subject matter, is not defined (e.g. some acronyms PLL, PI, ZCS, ZVS, MCU).
To read this journal a specific background is absolutely necessary, especially in the field of engineering. However, a good explanation of the concepts covered is mandatory.
I think it is very important to pay attention to these issues before submitting a paper.
Reviewer 3 Report
In this work, Moon et al described a novel resonance frequency tracking method that reduced the tracking time by two orders of magnitude, from tens of ms to hundreds of us. The work presented is of decent quality and the data is sufficient to support the claims. I have the following comments regarding this work:
1. Although novel, the impact of this proposed high-speed RFT method is unclear. What physical events that typically require the use of an ultrasonic system appear in below ms that require such high-speed RFT correction? What would be the impact of having a slower RFT in such situations? In many cases, the piezoelectric transducers can accommodate a certain range of applied frequencies without a notable change in output power. What types of ultrasound applications are most in need of this high-speed RFT?
2. The legend of the figures should be more specific. Many of them do not describe the content of the figure but only include a title. In figures 21-23 and 25-27, many parts of the figure are difficult to read.
3. In figures 3, 6 and 19, the x-axis of the figure is in weight instead of pressure, making the data in this figure sample size relevant. What are the sizes and characteristics of the piezoelectric materials used in generating this graph? In addition, a detailed description (method, instrument, materials) of how the experimental data were collected is missing.
Round 2
Reviewer 2 Report
In my opinion, the Authors have not sufficiently addressed my suggestions and have not made enough improvements. While the paper is really interesting, I think it is not well written and ready for publication.
